# Realization of wafer-scale nanogratings with sub-50 nm period through vacancy epitaxy

Qiushi Huang[1,2,8], Qi jia[1,3,8], Jiangtao Feng[2], Hao Huang[1,4], Xiaowei Yang[2,4], Joerg Grenzer[5], Kai Huang[1,3], Shibing Zhang[1,3], Jiajie Lin[1,3], Hongyan Zhou[1,3], Tiangui You[1], Wenjie Yu[1,3], Stefan Facsko [5], Philippe Jonnard [6], Meiyi Wu[6], Angelo Giglia[7], Zhong Zhang[2], Zhi Liu [1,4], Zhanshan Wang[2], Xi Wang[1,3] & Xin Ou[1,3]

Gratings, one of the most important energy dispersive devices, are the fundamental building blocks for the majority of optical and optoelectronic systems. The grating period is the key parameter that limits the dispersion and resolution of the system. With the rapid development of large X-ray science facilities, gratings with periodicities below 50 nm are in urgent need for the development of ultrahigh-resolution X-ray spectroscopy. However, the wafer-scale fabrication of nanogratings through conventional patterning methods is difficult. Herein, we report a maskless and high-throughput method to generate wafer-scale, multi-layer gratings with period in the sub-50 nm range. They are fabricated by a vacancy epitaxy process and coated with X-ray multilayers, which demonstrate extremely large angular dispersion at approximately 90 eV and 270 eV. The developed new method has great potential to produce ultrahigh line density multilayer gratings that can pave the way to cutting edge high-resolution spectroscopy and other X-ray applications.

[1] State Key Laboratory of Functional Materials for Informatics, Shanghai Institute of Microsystem and Information Technology, Chinese Academy of Sciences, Shanghai 200092, China. [2] Key Laboratory of Advanced Micro-Structured Materials MOE, Institute of Precision Optical Engineering, School of Physics Science and Engineering, Tongji University, Shanghai 200092, China. [3] Center of Materials Science and Optoelectronics Engineering, University of Chinese Academy of Sciences, Beijing 100049, China. [4] School of Physical Science and Technology, ShanghaiTech University, Shanghai 201210, China. [5] Institute of Ion Beam Physics and Materials Research, Helmholtz-Zentrum Dresden-Rossendorf, Bautzner Landstrasse 400, Dresden 01328, Germany. [6] Sorbonne Université, Faculté des Sciences et Ingénierie, UMR CNRS, Laboratoire de Chimie Physique – Matière et Rayonnement, boîte courrier 1140, 4 place Jussieu F-75252, Paris cedex 05, France. [7] CNR Istituto Officina Materiali, Trieste 34149, Italy. [8]These authors contributed equally: Qiushi Huang, Qi jia. Correspondence and requests for materials should be addressed to X.O. (email: ouxin@mail.sim.ac.cn)

The developments of modern information industry and modern scientific tools greatly depend on the advancements in nanofabrication processes that create nanometer size features on wafer-scale areas[1]. In the past decades, nanostructures were generally fabricated by top–down methods, such as conventional photolithography, which require masks that nearly reach their technical limits to further scale down the structure size[2]. While, another method for fabricating nanostructures by direct writing with electrons or ions is limited to small areas and is time consuming[3]. Nanoimprint lithography can generate well-defined nanostructures over large areas[4]. However, prefabricated templates are still required, and the overall process is complex. Alternatively, bottom-up methods based on self-assembly can provide a simple, robust, and rapid route for fabricating nanostructures economically. For example, the direct self-assembly of block copolymers has successfully generated well-defined sub-10-nm nanostructures with the macro-size area[5,6]. Low-energy ion irradiation of semiconductor surfaces can also generate nanoscale wave structures[7–9]. A large-area grating structure with a period down to 120 nm was reported to be fabricated through fracturing[10]. Nevertheless, fabricating regular nanostructures with a period below 50 nm on a wafer-scale area is still challenging, even when using state-of-the-art nanofabrication technology.

One-dimensional periodic nanostructures (nanogratings) are one of the most fundamental devices and have been widely applied in electronic, optical, magnetic, and biological applications[11–17]. One notable application is the diffraction grating in spectrometer, where the grating period determines the angular dispersion and resolution of the system. Extreme ultraviolet (EUV) and soft X-ray science have made tremendous progress in the past few decades, allowing the electronic and magnetic properties of complex solid-state systems to be understood due to its unique ability to probe inner shell electrons and study the excitation spectrum[18]. For the development of high-resolution spectroscopy techniques such as resonant inelastic X-ray scattering (RIXS), extremely high line density gratings with a large angular dispersion and good efficiency are urgently required[19]. Besides the high resolution, a compact spectrometer is demanded by the space-based telescope or a lab-based instrument, which also needs high-line density gratings. Current commercial gratings used for EUV and soft X-ray regions are generally fabricated by mechanical ruling, or laser interference lithography techniques and have a typical groove density of no more than 5000 lines mm$^{-1}$ (see refs. [20,21]). Even higher-line density, e.g., 10,000 lines mm$^{-1}$, blazed gratings have been demonstrated by e-beam lithography and the anisotropic etching of silicon single crystals[22]. However, it is still challenging to produce nanogratings with both high-groove density and large area.

In this work, we applied a maskless method to fabricate sub-50-nm periodic nanogratings with wafer-scale dimension by a vacancy epitaxy technique. To make the nanogratings function in the EUV and soft X-ray region, they are coated by a multilayer structure which enhances the diffraction efficiency through Bragg reflection. Wafer-scale sawtooth gratings with a groove density of around 20,000 lines mm$^{-1}$ and coated with Mo/Si and Cr/C multilayers were successfully fabricated. These two gratings were designed for operation in the energy range near the Si–L edge (99.8 eV) and C–K edge (284.2 eV), respectively. The diffraction efficiency and ultra-large angular dispersion of these multilayer gratings were demonstrated by EUV and soft X-ray measurements performed in a synchrotron radiation facility.

## Results

**Formation of nanograting pattern**. The self-assembly process of vacancy epitaxy is mitigated through low-energy ion irradiation.

As the surfaces of single-crystal semiconductors are irradiated by low-energy (less than 2 keV) ions at temperature above the recrystallization temperature, vacancies are created, which then self-assemble depending on the Ehrlich–Schwoebel barrier[23,24]. Diverse nanostructures can be formed based on the facet angles and symmetries of the crystal surfaces. In contrast to traditional epitaxy methods, such as molecular beam epitaxy (MBE), these nanostructures were self-organized vacancy by vacancy instead of atom by atom. Figure 1a shows the schematic for vacancy epitaxy process on GaAs(001) surfaces. When the GaAs (001) surfaces are irradiated at temperatures above ~350 °C, high-density surface vacancies are created. These vacancies diffuse according to the Ehrlich–Schwoebel barrier at the step edges. Due to the exclusive formation of dimer rows of Ga atoms, the symmetry on the (001) surfaces is broken[25]. Thus, the step edges aligned along the [1–10] and [110] directions are no longer energetically equal. With increasing ion-irradiation fluence, periodic, symmetric, faceted nano-grooves oriented along the [1–10] direction with a period of 48 nm can be formed, as shown in Fig. 1b, c. The 2D angle distributions in Fig. 1d also reveal that the facets on the irradiated surface have an angle with the GaAs(001) surface of ~19°, which is close to that of the [114] crystal plane. The surface roughness of the self-assembled facets is very small, only ~0.2 nm, as measured by AFM. This is smooth enough for the growth of X-ray multilayers. These sawtooth structures can be considered as building blocks of a blazed grating with symmetric facets.

Period and blaze angle are the key parameters for fabricating blazed gratings. As shown in Fig. 1e–g, the period of the nanogratings can be tuned by the irradiation temperature. With the ion-irradiation temperature increased from 450 °C to 480 °C or 520 °C, the period of the nanogratings increased from 48 nm to 55 nm or 78 nm, respectively. In addition, asymmetric facets are formed on the surfaces with predefined miscut angles as well. Figure 1h–j shows nanograting structures on vicinal GaAs(001) surfaces with miscut angles of 0°, 10°, and 15° in the [110] direction. The blaze/antiblaze angle of the formed nanograting structures changed from 19°/19° (Fig. 1h) on the GaAs (001) surface to 9°/30° (Fig. 1i) and 4°/35° (Fig. 1j) on the surface with 10° and 15° miscut angles, respectively. The area of the blazed facet increases with the miscut angle. However, the current asymmetric nanostructures are less regular and uniform than the symmetric nanostructures. Due to the tunable structural parameters, these self-organized sawtooth nanostructures are suitable for application as ultrahigh-line-density nanogratings.

**Design and fabrication of multilayer nanogratings**. Figure 2a shows the schematic of the sawtooth structure, which is similar to that of a blazed grating. Given the extremely high line density and the angular dispersion, the grazing incidence angle with respect to the surface of the facets under working conditions of a blazed grating will exceed the critical angle of any material, including gold in the EUV and soft X-ray range. Thus, single-layer-coated nanogratings have a very low efficiency. Multilayer coatings are highly efficient optics in the short-wavelength region (Fig. 2b), due to the constructive interference of multiple reflections from many interfaces[26]. Multilayers with an optimized structure deposited on top of the nanogratings to form a 3D-diffraction structure endows the ultrahigh-line-density gratings with high efficiency (Fig. 2c)[27–29]. To achieve the resonant condition for maximum efficiency, the line density, blazed angle, and the d-spacing of the multilayer need to be matched, in order to satisfy both the grating equation condition and the generalized Bragg condition[27]. Generally, working at higher photon energy requires a smaller blaze angle of the multilayer nanograting with a fixed line density. Meanwhile, a smaller blazed angle leads to smaller d-

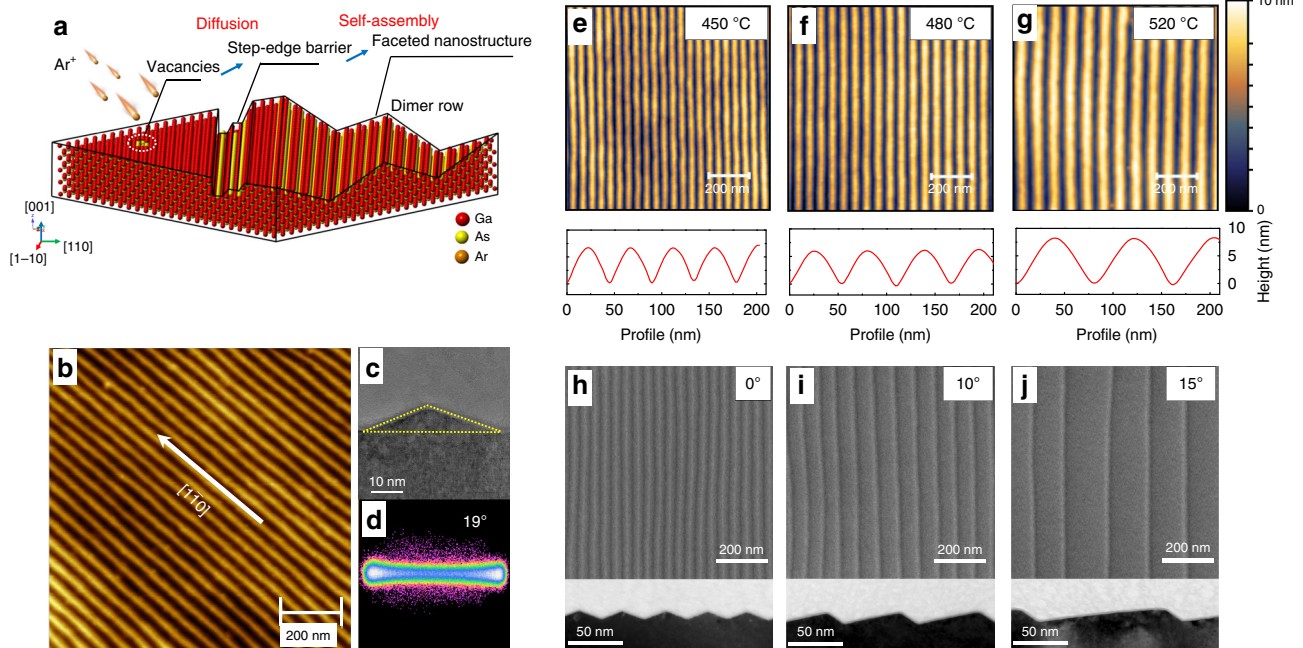

**Fig. 1** Fabrication of the nanograting substrates. **a** Schematic of the vacancy epitaxy on GaAs(001) surfaces. **b**, **c** AFM and XTEM images of the sawtooth GaAs substrate with a nanogroove density over 20,000 lines mm$^{-1}$ and (**d**) distribution of the facet angles over the AFM image, mostly 19°. **e–g** AFM images of the GaAs (001) substrates and their cross-section profiles along the [110] direction irradiated at 450 °C, 480 °C, and 520 °C. The periodicities of the nanogratings are 48 nm (**e**), 55 nm (**f**), and 78 nm (**g**), respectively. **h–j** SEM and XTEM images of the nanogroove structures formed on GaAs (001) with a miscut angle of 0°, 10°, and 15° toward the [110] direction. The periodicities of the nanogratings are 48 nm (**h**), 55 nm (**i**), and 155 nm (**j**), respectively

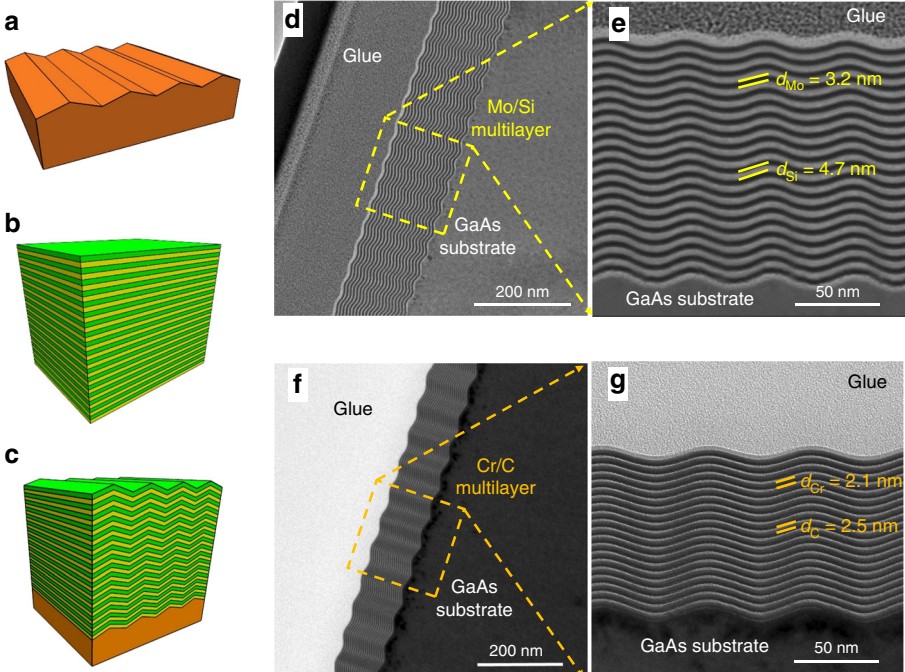

**Fig. 2** Fabrication of multilayer blazed gratings. Schematic of the (**a**) bare nanogrooved substrate, (**b**) multilayer, and (**c**) multilayer grating. Cross-section TEM images of the nanograting coated with Mo/Si multilayers (**d**, **e**) and Cr/C multilayers (**f**, **g**)

spacing of the multilayer. So the blaze angle cannot be too small in order to make the d-spacing achievable. Using a higher-order diffraction allows the use of a larger blaze angle than the 1st order would require. One method to reduce the initial blaze angle while keeping the grating period unchanged is to coat the grating with a thick layer to smooth out the grating profile. This smoothing effect will be seen in the following deposition results of the multilayers. Due to the extremely small period of the grating structure, the highest working energy is limited compared with the conventional multilayer gratings with larger period[30], according to the above-mentioned principles. For instance, assuming a smallest d-spacing of 3 nm and the grating period of

50 nm, the highest working energy is below 3 keV. A detailed diffraction model of the ultrahigh line-density grating is being built and will be presented in another work. Two multilayer gratings were designed to operate in the EUV and soft X-ray range (for details see Supplementary Fig. 1, Supplementary Tables 1, 2, and Supplementary Note 1). The first grating was a Mo/Si multilayer grating with a grating period of 50 nm, a multilayer d-spacing of 8.1 nm, and thickness ratio of Mo of 0.4, which met the resonant condition of the +2nd diffraction order at ~90 eV. The theoretical diffraction efficiency at a photon energy of 89.2 eV with 20 pairs (number of bilayers) Mo/Si layers was 32% at an incident angle of 5°. The second grating was a Cr/C multilayer grating with a grating period of 60 nm, d-spacing of 4.8 nm (thickness ratio of Cr of 0.4) and 20 bilayers, which functioned at the −4th diffraction order at ~270 eV. Using the −4th high order can further improve the angular dispersion and keep the optimal blaze angle still as 19°. The theoretical −4th order efficiency reaches 6% at the photon energy of 263.5 eV and the incident angle of 75°. The lower efficiency at higher energy can be due to the effect of the symmetric nanograting structures working at grazing incidence geometry. A relatively small number of bilayers ($N = 20$) was used for the first demonstration to reduce the smoothing of the grating facets during multilayer deposition[31]. The theoretical efficiency can be much improved by changing the grating groove into an asymmetric shape, close to the ideal blazed grating, via the fabrication with a miscut GaAs crystal as mentioned above. Alternatively, if the multilayer can be deposited mainly on one side of the grating facets (blaze facets) so that the absorption effect from the multilayer on the other side of the facets (antiblaze facets) is canceled, the efficiency will also significantly increases up to almost the same value of the ideal blazed grating[27]. This can be explored by using the glancing deposition technique instead of the normal incidence deposition currently used.

Figure 2d, e displays the XTEM images of these two multilayer gratings. The first multilayer-blazed grating (MBG) has a grating period of ~49 nm and was coated with a 20 pairs Mo/Si multilayer. The thickness of each Mo and Si layer was 3.2 nm and 4.7 nm, respectively. The second MBG has a grating period of ~62 nm and was coated with a 20 pairs Cr/C multilayer. The thickness of each Cr and C layer was ~2.1 nm and 2.5 nm, respectively. As shown in the images, the grating profile was mostly replicated by the layers forming a 3D nanograting structure with high regularity, which is crucial in order to achieve a reasonable efficiency. Although the fabricated grating profiles are close to the sinusoidal shape rather than triangular ones, the theoretical efficiency of the multilayer nanograting with the real substrate profile obtained from the XTEM image is only a little smaller than the triangular one according to the simulations. This sinusoidal-like shape can be further taken into account in the design. Both the two multilayers are grown with sharp and smooth interfaces on the sawtooth structures, which indicate the high quality of the layers. A slight smoothening of the grating profile was also observed from the bottom to the top of both multilayer gratings. This will be further discussed in the following section.

**Characterization of the diffraction efficiency and angular dispersion**. The diffraction performance of these two MBGs were measured at the BEAR beamline at Elettra Synchrotron in Trieste. For the Mo/Si multilayer grating, the incident angle was set to 5° from the grating surface normal (Fig. 3a). A wide-range detector scan was performed to measure the diffraction intensity of different orders at the photon energy of 87.5 eV. As shown in Fig. 3c, the +1st order has the maximum peak efficiency given to

the satisfaction of its resonant diffraction conditions at this incident angle and energy, while the neighboring orders are around two orders of magnitude lower. According to the very large angular separation of the 0th, +1st, and +2nd orders, the grating period is calculated to be 48.6 nm, which is consistent with the XTEM measurement. Due to the grating dispersion and the small variation of the grating period, the angular widths of the diffraction peaks are larger than the incident beam. Thus, to estimate the integral efficiency of a single order, a larger detector slit was used to collect the whole beam. By fixing the detector angle at the +1st and +2nd-order positions, the variation in integral efficiency versus the photon energy was measured, as shown in Fig. 3d. The maximum integral efficiency of the +1st order is 11% at 87.5 eV, while the +2nd order displays a lower value of 7% at 93.8 eV. This is different from the designed result of the grating structure with a blaze angle of 19°, in which the +2nd order should have the highest efficiency. By using the real multilayer grating structure with gradually smoothed profile measured by XTEM (Fig. 3a), the theoretical efficiency was calculated as shown in Fig. 3d. The 1st order indeed showed a higher efficiency of 25% than the 2nd order, and the peak positions were close to the measured curves. To further study the effect of the smoothed grating profile on efficiency, the blaze angles after different number of bilayers deposited were extracted from the TEM image (Fig. 3e). The theoretical efficiency of the Mo/Si multilayer grating with the reduced blaze angles were calculated and shown in Fig. 3f. As the blaze angle decreases from 18.65° (bottom area) to 14.27° (middle area) and 10.82° (top area), the 2nd order efficiency drops continuously while the 1st order efficiency increases. Evidently, the grating with the smaller blaze angle is more matched with the multilayer structure at the 1st order than the 2nd order which transfers the energy mainly to the 1st order. The reference Mo/Si multilayer with the same structure was also measured which showed a reflectance of 46% at 90 eV, under an off-normal incidence angle of 20°. Assuming the multilayer imperfections have the same effect on multilayer reflectance and grating efficiency, and using the real groove shape after the coating, the expected efficiency of the Mo/Si multilayer grating is ~18%. In this case, the experimental efficiency of 11% is only a little lower than the expected value, which indicates a relatively good quality of the nanograting. The remained discrepancy between the simulated and experimental efficiency can be explained by the imperfections of the nanograting structure and the slightly different layer structure grown on the flat substrate and on the sawtooth structures.

The diffraction efficiency of the Cr/C multilayer grating working at near the carbon K edge was measured in the same manner. Due to the high photon energy, the incidence angle was fixed at 75° from the grating normal, and the efficiency distribution among different orders was measured at 271.4 eV, as shown in Fig. 3g. The −4th order showed the highest peak efficiency, and the neighboring orders were one order of magnitude lower. The grating period calculated from the diffraction angles was 60 nm, which is also close to the XTEM result. The integral efficiency of the −4th order is displayed in Fig. 3h, and the maximum value was 1.2% at 271.4 eV. For comparison, the theoretical curve calculated using the real grating profile from XTEM is shown, which also has a low efficiency of only 2.9% compared to the ideal structure due to the smoothed profile.

To evaluate the angular dispersion of the optics, two-dimensional scans of both the photon energy and diffraction angle were performed under a fixed incidence. For the Mo/Si and Cr/C multilayer gratings, the dispersion was measured around the resonant conditions of the +1st and −4th order, respectively, as determined above. As shown in Fig. 4b, c, the central high

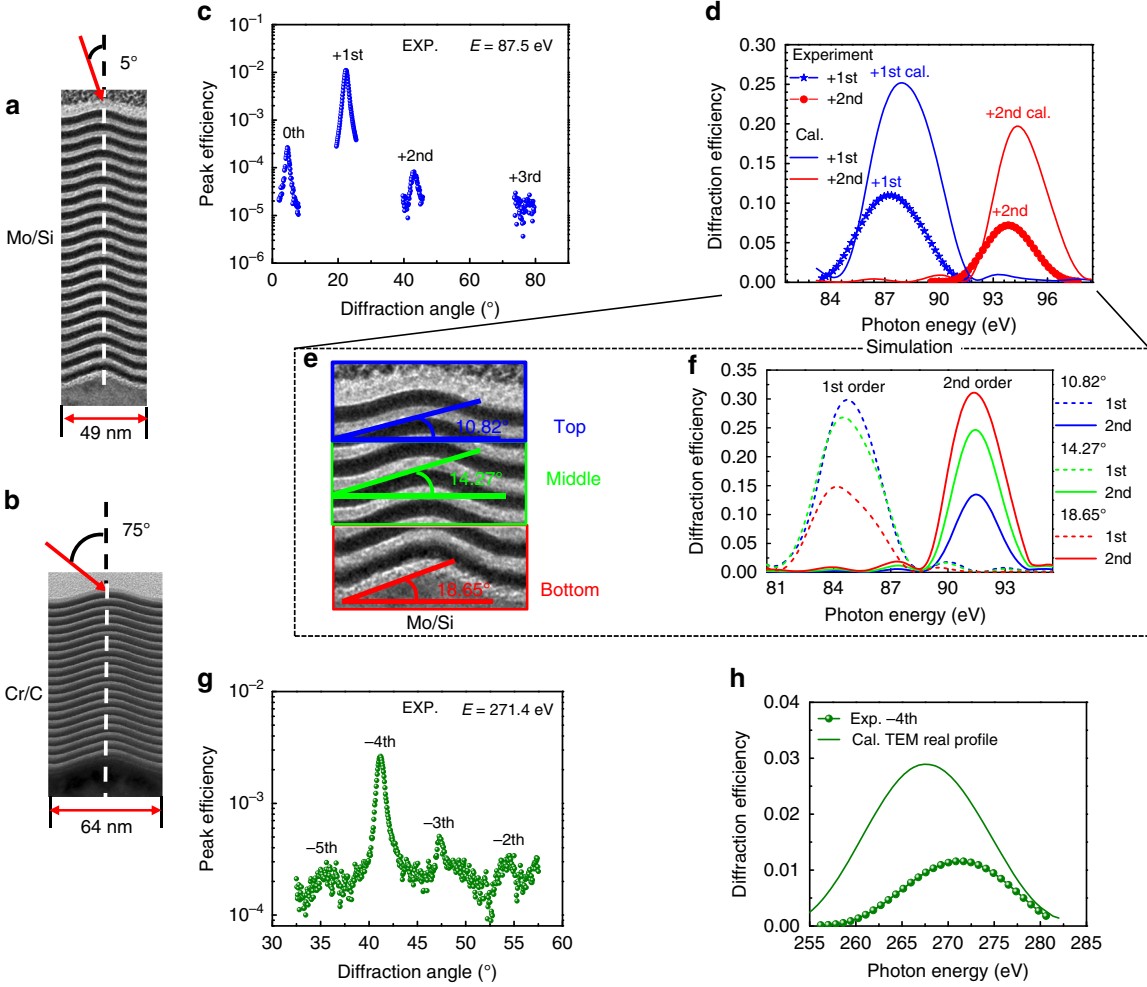

**Fig. 3** EUV and soft X-ray measurements of the nanogratings. Schematic of the Mo/Si (**a**) and Cr/C (**b**) MBGs working at incident angles of 5° and 75°, respectively. Efficiency distribution (detector scan) of the Mo/Si (**c**) and Cr/C (**g**) MBGs were measured at a photon energy of 87.5 eV and 271.4 eV, respectively. The experimental (stars and circles) and simulated (solid lines) curves of the integral diffraction efficiency of the 1st (blue lines) and 2nd (red lines) orders of Mo/Si MBG are shown in (**d**). The experimental (spheres) and simulated (solid lines) curves of the integral diffraction efficiency of the −4th order of Cr/C MBG is shown in (**h**). The simulations in (**d**) and (**h**) were performed with the real structures derived from the XTEM images shown in (**a**, **b**). The local groove profiles at the top, middle, and bottom positions of the Mo/Si BMG stack are shown in (**e**) and the simulated efficiency of the 1st and 2nd orders of Mo/Si BMGs using the three different groove profiles are shown in (**f**)

efficiency area is attributed to the enhancement of the multilayer Bragg diffraction due to the satisfaction of both the generalized Bragg condition and grating equation. The side lobes in Fig. 4b are caused by the multilayer interference effect. For both multilayer gratings, the angular position for peak efficiency at each energy was determined and is connected by a dashed line within the central high efficiency area in the figure. The slope of this line equals $\Delta\theta/\Delta E$, representing the angular dispersion of the optics. For the Mo/Si multilayer grating measured at 5° incidence, the angular dispersion was ~0.21°/eV in the energy range of 81–97 eV. This is consistent with the theoretical value of a nanograting with a 48.6 nm period, derived by the grating equation. For the Cr/C multilayer grating measured at 75° incidence using the −4th diffraction order, the angular dispersion was ~0.093°/eV in the energy range of 262–278 eV. This is also in consistent with the theoretical dispersion of the grating with 60 nm period. The demonstrated dispersion of the fabricated multilayer gratings is 4.5–6.3 times larger than a conventional 5000 lines mm⁻¹ grating at ~90 eV and 270 eV, respectively. The line density of 5000 lines mm⁻¹ is almost the highest value achievable by the modern ruling technique[20]. Assuming the other

technical parameters, like the variation of grating period, slope error of the grating substrate, spectrometer set up etc., are the same, the essentially larger angular dispersion can easily separate two adjacent wavelengths on the image plane and provide a much higher spectral resolution than the conventional 5000 lines mm⁻¹ grating. On the other hand, the ultrahigh angular dispersion allows for a compact design of the spectrometer, affording a much smaller instrument with the same resolution as currently achieved. It should be noted that the assumption above of identical technical parameters of the new nanogratings as the modern grating optics is not fully valid yet. Especially for the broad intensity distribution of the measured diffraction peak that is an important limiting factor for the spectral resolution despite the large angular dispersion[32]. As seen in Fig. 4c, the angular width (full-width half-maximum, FWHM) of the diffraction peak at the center resonant energy is much larger than the width of the direct beam. After subtracting the broadening effect from the instrumental factors and the expansion diffraction geometry, the variation of the grating period was found to be the dominant reason. The relative variation of the grating period was estimated to be ~± 2%, which equals to an absolute error of ±1 nm. This is

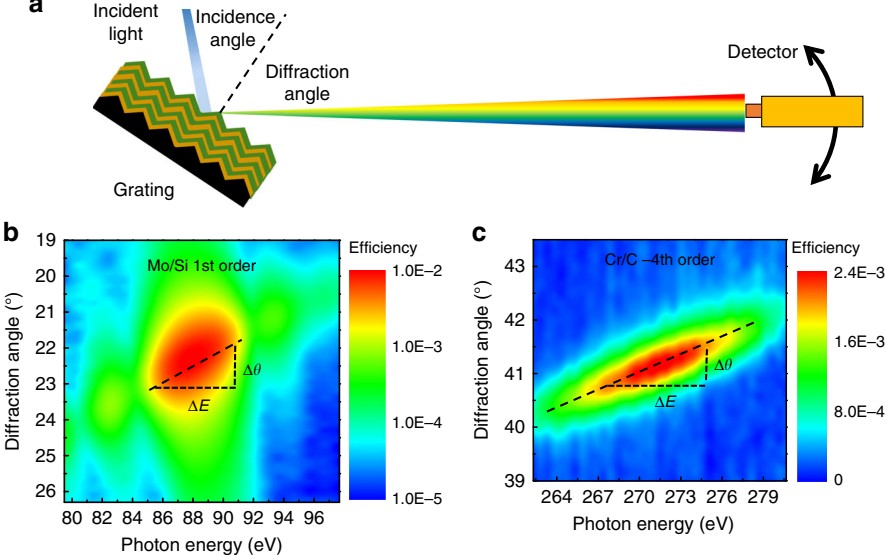

**Fig. 4** Estimation of the angular dispersion of the fabricated MBGs. **a** Schematic of the diffraction geometry of the gratings. **b**, **c** Two-dimensional diffraction measurements as a function of diffraction angle and photon energy of the +1st order of the Mo/Si MBG at an incident angle of 5° and the −4th order of the Cr/C MBG at an incident angle of 75°, respectively

in consistent with the value analyzed from the AFM measurements which is ± 1.2% ~± 2.3%. The small period error still needs to be reduced to eventually achieve the ultrahigh resolution.

**Multilayer nanograting on the wafer scale**. For application in an X-ray spectrometer, a high-quality large-size grating is needed to increase the angular acceptance and sensitivity in real experiments. The resolving power of a grating is also determined by the total number of grooves illuminated on the grating (i.e., the grating aperture). To demonstrate the large-size fabrication capacity of this technique, a 2-inch, wafer-scale sawtooth nanograting was produced using a broad ion-irradiation beam (Fig. 5a). In principle, the grating area depends on the size of the ion beam, which can be increased without physical limit. This represents a unique advantage of this technique. Grazing incidence small-angle X-ray scattering (GISAXS) analysis was performed to measure the lateral ordering and orientation of the nanostructure on the wafer-scale sample. The beam size on the sample is 0.4 mm (width) × 10 mm (projected length). Figure 5b shows the GISAXS pattern of the nanograting over which the X-ray beam is parallel to the grating groove direction (GaAs[1–10]; phi = 90°) and the integrated line profile along the in-plane scattering vector $q_∥$. All grating rods of the GISAXS patterns are well separated and distinct. The period of the nanograting was estimated as 41 nm with an uncertainty error of ± 2%. Taking into account the instrumental factors that can also lead to the period error, the period variation of the grating within the illuminated area should be <± 2%, which is consistent with the AFM and diffraction peak analysis results above. Due to the scattering geometry and source intensity limitations, the GISAXS pattern measured with the X-ray beam perpendicular to the groove direction (phi = 0°) did not show the grating peaks. The uniformity of the wafer-scale grating was investigated by SEM measurements at seven different sites on the wafer (Fig. 5a). The periods determined from each SEM image are in the range of 40.8 ± 0.6 nm, which shows a relatively good uniformity of the nanograting structures (Fig. 5c). To further verify the consistency of the diffraction property over the whole grating, a 20 pairs Mo/Si multilayer was deposited on this wafer-scale substrate. Figure 5d shows the diffraction results of the nanograting

measured at a photon energy of 98.7 eV and an incidence angle of 20°. The grating was measured at the center, middle, and edge positions along the radial direction of the wafer, as shown in the inset of Fig. 5d. The three diffraction curves are very close to each other, especially for the efficiency of the resonant −1st order peak. According to the slight difference of the peak positions of the −1st order, it was estimated that the grating period gradually increased from the center area to the edge by about ± 2.1%, which is similar to the error found by SEM. The gradual change of the period can be caused by the non-uniformity of the ion beam from the 2-inch source and the slight temperature gradient of the heating plate in the vacancy epitaxy process. These can be improved by using a larger ion source and a more uniform heating plate that are commercially available.

**Discussion**
The two MBGs developed in this work are the first demonstrations of such nanogratings. Further improvement can be made in the diffraction efficiency and period uniformity for high-resolution applications. The working energy region can also be extended. First, the efficiency of the symmetric nanogratings can be improved by using asymmetric nanogroove structures self-organized on vicinal GaAs (001) surfaces with miscut angles as grating substrates, which provides a better groove profile close to the ideal blazed grating. Alternatively, the glancing deposition technique can be explored to deposit multilayer mainly on one side of the symmetric facets, which can also significantly enhance the diffraction efficiency. Smoothing of the groove profile during multilayer deposition induces a deviation of the real grating (layer) profile from the optimum one and thus reduces the efficiency. The smoothing effect should be taken into account in the design and a dedicated deposition process for conformal layer growth on the nanostructures also needs to be developed[33]. Second, high-resolution spectroscopy requires a precise control of the period uniformity with an error below ± 0.5%. For a 20,000 lines mm$^{-1}$ grating, it means an absolute error of <± 0.25 nm, which is a challenge for every existing grating fabrication technique. To achieve this, an in-depth study of the mechanism of the vacancy epitaxy process and systematic engineering works are needed. The latter one includes the high-quality GaAs substrate

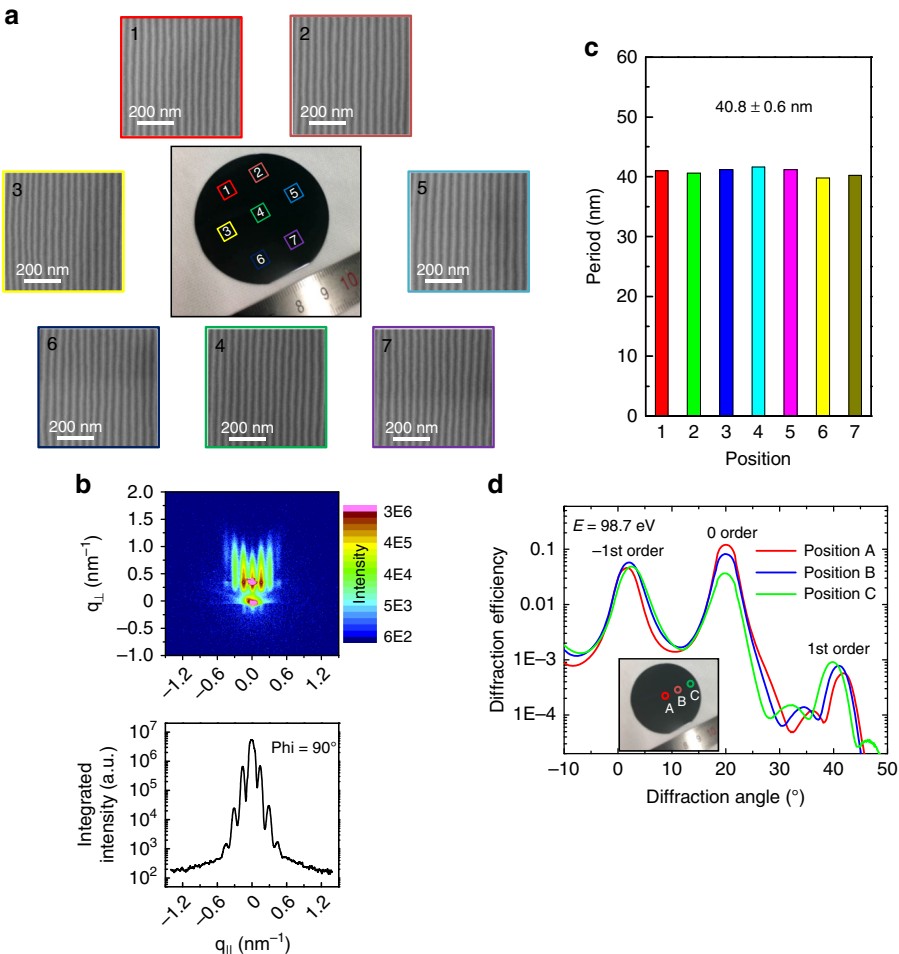

**Fig. 5** Uniformity characterization of the 2-inch wafer-scale nanograting. **a** Optical microscopy image of the 2-inch wafer-scale nanograting and SEM images of the nanostructures at different positions on the wafer, (**b**) GISAXS pattern and the integrated line profile of the nanograting structure over which the X-ray beam is parallel to the direction of the groove GaAs[1−10] (phi = 90°) ($q_{||}$ is the in-plane scattering vector.), (**c**) the grating periods determined from the SEM images, and (**d**) diffraction efficiency measurements of the Mo/Si multilayer-coated nanograting at a photon energy of 98.7 eV at three different positions on the wafer-scale grating

in terms of minimizing lattice defects/impurities and better surface figure, optimized ion irradiation parameters, better homogeneity of the ion beam and the temperature of the irradiation area, and so on. The line density of the nanograting can also be tuned from ~22,000 lines mm$^{-1}$ to 12,000 lines mm$^{-1}$, which provides a parameter window to balance the resolution and photon flux for different applications. Third, the multilayer nanogratings developed in this work mainly work at the low-energy region. For the high-energy region, e.g., up to 1 keV, other multilayer systems with smaller d-spacings are needed. The blaze angles of the grating also need to be reduced to below 19° as defined by the crystal geometry, down to ~7°–8°. With these improvements, we believe that this new concept and method will be able to provide ultrahigh line density gratings with a large size that can realize higher resolution than currently achieved for advanced soft X-ray spectroscopy applications.

Moreover, the developed nanogratings can also be used for other applications where a large angular dispersion is more important than the high resolution and high precision of the structures. For instance, the nanograting can be used for the EUV and soft X-ray astronomical observation. The large dispersion of the grating is important for the compactness and light weight of the spectrometer, which can significantly reduce the cost and enable the observation through a small satellite platform. Similar

compact spectrometer systems are also widely needed in lab-based instruments. Besides the application in a spectrometer, the symmetric nanograting working at the conical diffraction mode can be used as a beam splitter, which generates two diffracted beams (±1st orders) with the same intensity and a transmitted beam (0th order) simultaneously. The large angular dispersion will provide a large separation of the split beam. This is very useful for photon diagnostics and pump–probe experiments applied in the free electron laser facilities[34]. Another application is the key dispersion optics in an X-ray pulse shaper system which can further compress the FEL pulse into sub-femtosecond regime and fully control its spectro-temporal properties[35,36]. The structures of such pulse shaper have been reported which typically require two gratings[29,35,36]. A large angular dispersion enables the compensation of a large chirp, an accurate control of the phase characteristics, and a compact system.

In summary, we successfully demonstrated a new method that can be used for the fabrication of high line-density multilayer gratings with over 20,000 lines mm$^{-1}$ working at the EUV and soft X-ray region. These gratings were generated on GaAs (001) surfaces by "vacancy epitaxy" process via broad low-energy Ar ion irradiation at elevated temperature. The period and facet angles can be tuned by adjusting the irradiation temperature and the miscut angles of the substrate. Through combination with a

high-reflectance multilayer and forming a 3D diffraction structure, a diffraction efficiency of 11% at the 1st order was obtained at a photon energy of 87.5 eV. Using a higher diffraction order, e.g., the 4th order, enabled by Bragg diffraction, the angular dispersion was further enhanced, with a diffraction efficiency of 1.2% measured at 270 eV. The measured angular dispersion of the multilayer nanogratings is 4.5–6.3 times larger than a conventional 5000 lines mm$^{-1}$ grating, which is almost the highest density achievable by modern fabrication technique. A 2-inch wafer-scale nanograting was also demonstrated, proving the unique advantage to produce large-area nanograting structures in a single step with relatively good uniformity. With further improvement of the structure quality, these ultrahigh-line-density multilayer gratings can be used to promote the development of ultrahigh-resolution spectroscopy. It also represents a step toward the development of a compact X-ray spectrometer, a beam splitter, or an ultrashort X-ray pulse shaper with high performance.

## Methods

**Fabrication of the nanogratings.** The nanograting substrates were fabricated on GaAs (001) wafers by the normal incidence of 1 keV Ar$^+$ irradiation at elevated temperature (400–500 °C) with an ion fluence of $1 \times 10^{19}$ cm$^{-2}$. All irradiation processes were performed under vacuum at a pressure of $10^{-8}$ mbar. The topographies of the nanogratings were characterized by a Bruker multimode-8 atomic force microscope in tapping mode and a JSM-7800F scanning electron microscope. The multilayers were fabricated by direct current magnetron sputtering technique. The base pressure before deposition was around $2 \times 10^{-6}$ mbar. High-purity argon (99.999%) was used as the working gas at a pressure of $2 \times 10^{-3}$ mbar. The deposition rates of Mo, Si, Cr, and C were 0.08 nm s$^{-1}$, 0.09 nm s$^{-1}$, 0.09 nm s$^{-1}$, and 0.03 nm s$^{-1}$, respectively. The microstructure of the Cr/C multilayer with a similar d-spacing was studied in our former work[37]. During deposition of the multilayer gratings, witness samples were deposited on super-polished Si wafers that had the same structure as the multilayer on the gratings. The witness samples were measured by hard X-ray reflectometry at $\lambda = 0.154$ nm. Based on the fitting results, the fabricated structures of the Mo/Si and Cr/C multilayers were determined. Detailed structure information of the multilayers was further obtained using a JEM-2100F transmission electron microscope. The GISAX measurements were carried out at an Empyrean diffractometer, at the same wavelength, using—on the source size—a point focus and a side-by-side multilayer optics. A Medipix detector was used to collect the scattered signal from the multilayer grating.

**Diffraction measurements.** The diffraction efficiency and dispersion of the multilayer gratings were characterized at the Bending magnet for Emission, Absorption, and Reflectivity (BEAR) beamline at the Elettra synchrotron. A 1200 lines mm$^{-1}$ grating was used in the monochromator of the beamline, and the exit slit was set to a width of 100 μm, which provides energy bandwidths of ~0.04 eV at 84–100 eV and 0.2 eV at 255–285 eV. The incident beam divergence was +/−1.0 mrad with a degree of s-polarization light of >96%. For the integral efficiency measurement, a large slit with a width of 8 mm was inserted in front of the photodiode with an acceptance of 3° to collect the whole beam. For the measurements of efficiency distribution among neighboring orders and dispersion measurements, a small slit with a width of 0.5 mm and 0.18° acceptance was used to improve the angular resolution. The efficiency was determined by normalizing the measured intensity to the direct incident beam using the same slit. For the diffraction measurements of the wafer-scale grating, the large slit was used.

## Data availability

Data supporting the findings of this study are available from the corresponding author on request.

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

## Acknowledgements

This work was supported by the National Natural Science Foundation of China (Nos.: U1732268, 11227902, 11622545, 61851406, 61874128, and 61621001), National Key R&D Program of China (Nos. 2016YFA0401304, 2017YFA0403302), Shanghai Science and Technology Innovation Action Plan Program (Grant no. 17511106202), Frontier Science Key Program of CAS (No.: QYZDY-SSW-JSC032), and One Hundred Talent Program of CAS, International Collaboration Project of Shanghai (No.: 16520721100). We are grateful to the enlightened suggestions and discussions provided by Dr. Igor V. Kozhevnikov (Shubnikov Institute of Crystallography of Federal Scientific Research Centre "Crystallography and Photonics" of the Russian Academy of Sciences") and Dr. Andrey Sokolov (Helmholtz-Zentrum Berlin für Materialien und Energie, BESSY-II).

## Author contributions

X.O., Z.W., Z.L., S.F., and X.W. supervised the project. Q.J., X.O., H.H., K.H., S.Z., J.L., H.Z., T.Y., and W.Y. performed the fabrication of nanograting structure. Q.H., J.F., Z.Z., and X.Y. performed the multilayer growth. Q.H., Q.J., J.G., P.J., M.W., and A.G. performed the diffraction measurements of the nanograting. Q.J., Q.H., and X.O. co-wrote the paper. All authors discussed and commented on the paper.

## Additional information

**Competing interests:** The authors declare no competing interests.

