## [Peer Review File · Nature Communications]

Review of manuscript

“Realization of wafer-scale ultrahigh-resolution nanogratings with sub-50 nm periodicity through vacancy epitaxy”

Authors of the manuscript propose a new method of making x-ray diffraction gratings having high groove density and high diffraction efficiency. Such gratings are of great importance for high resolution x-ray spectroscopy applications. The authors aimed to use a cost effective and high throughput bottom-up process for making a blazed grating and then coat it with a multilayer for high diffraction efficiency. The approach is based on two key technologies which are an ion beam induced faceting of a crystal surface to obtain periodical nano-patterns over a large area, and deposition of an x-ray multilayer. Both process are well known for decades while combination of both of them for making multilayer blazed gratings has never been investigated so far.

The authors made two multilayer coated gratings and performed their comprehensive characterization. Authors came to a conclusion that they “successfully demonstrated a new method for the fabrication of ultrahigh-resolution EUV and soft X-ray diffraction gratings with a line density of over 20,000 lines/mm.” This seems somewhat premature though. In fact the authors’ results show that the blazed gratings made by the method proposed by the authors are of poor quality and cannot be used for high resolution x-ray spectroscopy applications. For example, AFM and TEM data show that groove profile of the gratings obtained from the (100) crystals is sinusoidal rather than triangular. Although much better blazed profile was obtained for the miscut crystals, those gratings were not coated by a multilayer and their performances were not investigated. The miscut gratings probably were ruled out from further investigation due to substantial high-frequency errors in periodicity. High-frequency errors are much smaller for (100) crystals but they still are of significant portion of the grating period as seen on SEM and AFM images.

Authors claim that “the diffraction efficiency and *ultrahigh-resolution* of these multilayer gratings were demonstrated.” However, GISAXS measurements showed +/-5% low-frequency variations of the grating period (Fig. 5c). This is not a small number for high resolution applications. For example, gratings for high resolution spectrometers (such as one in Ref. [31,32]) often have variable groove density which gradually changes according to a specific law by +/-5% or so over the whole grating length. The random low frequency variations of the groove density should be at least an order of magnitude smaller (say 0.5%) otherwise the imaging properties and hence the resolving power of the spectrometer will be compromised. Moreover, the reported 5% errors in groove period were extracted from the GISAXS measurements performed along the grooves, when a relatively small number of the grooves was illuminated (due to a small beam footprint in the transverse direction). It would be more correct to evaluate the periodicity errors from the $q_{\text{perpendicular}}$ data for $\phi=0^\circ$ geometry (Fig. 5d). Since all the grooves are illuminated in this geometry the errors might be found much higher than 5%. In the view of the mentioned above the authors’ statements such as “we successfully demonstrated a new method for the fabrication of *ultrahigh-resolution* EUV and soft X-ray

diffraction gratings” or “the diffraction efficiency and *ultrahigh-resolution* of these multilayer gratings were demonstrated“ seem not supported by the results reported.

The poor shape of the grating grooves and high-frequency errors in grating periodicity are consistent to the low diffraction efficiency of the multilayer coated gratings. For example, diffraction efficiency of about 10% was measured for the Mo/Si coated grating while efficiency of an ideal grating is expected to be close to the reflectance of the multilayer which is about 60% (according to http://henke.lbl.gov/optical_constants/). Even more drastic mismatch between efficiency of 1% (Fig. 3f) and expected reflectance of about 15% of a Cr/C multilayer is observed.

Although the manuscript demonstrates that the proposed method is currently incapable to provide high performance of the gratings, this does not mean that the approach will never work. On the contrary, the bottom-up approach seems very promising provided the issues revealed by the authors are addressed. Moreover, the method might work “as is” for applications which are less demanding in terms of precision requirements as the x-ray gratings are. The value of the work is evaluation of the self-organization process in terms of production of x-ray gratings. This is important initial step which shows directions of further developments and investigations, and will be definitely interesting for the x-ray optics audience. I do not find this manuscript suitable for publication in Nature Communications since it does not contain a significant breakthrough in science or technology. I would recommend publication of this manuscript in more technically oriented journals like Applied Optics or Optics Express.

Reviewer #2 (Remarks to the Author):

This is very nice work that using vacancy epitaxy to create sub-50 nm gratings for soft X-ray optics applications. The method is novel and the work appears to be quite comprehensive, including the fabrication parameter space exploration and optical grating characterization. I recommend accepting for publication with the following minor changes:

- this appears to be the first report on vacancy epitaxy. If so, atomic resolution TEM images to show the vacancy formation and reorganization should be provided in order to confirm the mechanism.
- surface smoothness: the authors place that part in the supplemental information. I would recommend moving that to the main text, because it is important to address surface scattering for optical gratings.
- the multilayer Mo/Si and Cr/C deposition condition on top of the GaAs grating structure needs to be described clearly and the effect of the GaAs surface smoothness on the layers above need to be mentioned.
- in terms of the impact of this work, what is the fundamental limit of the range of grating periodicity using this method? How general is this method? Other than GaAs, would this method apply to other compound semiconductors with zinc-blende structure with aligned dimer surface? Since this is going to be used to soft X-ray optics, is GaAs hardy enough for extended exposure?

Reviewer #3 (Remarks to the Author):

The authors present a promising approach for manufacturing grating with ultimate density groove up to 20'000 lines/mm. The authors fabricate sub-50 nm periodic grating by vacancy epitaxy by low energy ion irradiation and multilayer structure growth on those nano grating to improve transmission by high order Bragg diffraction. This approach is in principle not new but this manuscript clearly shows the potential of this approach for future ultimate groove density grating.

Beside the high value of the presented work, I am some comments.

- 1) The blazed grading are important to improve the transmission of the grating by selecting the appropriate blazed angle. In the presented work it is not clear how the blazed can be selected independently from the groove density. Some comments of the coupling between the blazed angle and groove density will be welcome to better understand the strength and/or weakness of this approach with respect other methods.
- 2) The blazed grating quality are defined with the blazed angle (exposed surface) and "anti" blazed angle (unexposed surface) which are in principle highly asymmetric. It seems that the ultimate groove density grating (55nm) shows a symmetric structure which indicates the limitation of the applied method. Some comments on this limitation are requested.
- 3) The claimed energy resolution is based on a linear fit between photon energy and diffraction angle (see figure 4b and 4c) extrapolated to a 11 meter spectrometer. This is in no means a good measurement/estimation of the ultrahigh energy resolution of the grating. We strongly recommend to rephrase this part of the manuscript including title. Moreover no comments are provided regarding the "width" of the intensity distribution (in figure 4 b and 4c) which is a clear limiting factor in the true energy resolution. Same applied on optical quality of the blank substrate (for example slop errors) which will limit the energy resolution. Some open question remains also on the remaining optical quality of the blank after the low energy ion irradiation in the proposed approach.
- 4) Figure 4b shows interesting features such as "side lobes" which are not explained. Moreover those "side lobes" are mainly dominating the linear fitting between photon energy and diffraction angle, therefore it is mandatory to give a physical interpretation of those features. In addition, the central feature of Figure 4b shows a different dependence between photon energy and diffraction

angle that the one define by the "side lobes". This request a more detail understanding of the observed features.

In conclusion, I could recommend this manuscript only after major corrections

Reviewer1:

- 1. The authors made two multilayer coated gratings and performed their comprehensive characterization. Authors came to a conclusion that they “successfully demonstrated a new method for the fabrication of ultrahigh-resolution EUV and soft X-ray diffraction gratings with a line density of over 20,000 lines/mm.” This seems somewhat premature though. In fact the authors’ results show that the blazed gratings made by the method proposed by the authors are of poor quality and cannot be used for high resolution x-ray spectroscopy applications. For example, AFM and TEM data show that groove profile of the gratings obtained from the (100) crystals is sinusoidal rather than triangular. Although much better blazed profile was obtained for the miscut crystals, those gratings were not coated by a multilayer and their performances were not investigated. The miscut gratings probably were ruled out from further investigation due to substantial high-frequency errors in periodicity. High-frequency errors are much smaller for (100) crystals but they still are of significant portion of the grating period as seen on SEM and AFM images.**

Response: Thanks for the reviewer’s comments. We agree with the reviewer that this work shows a proof of principle of a novel optics that is very promising for making high performance X-ray gratings. Although the structure quality needs further improvement for certain demanding application like the high resolution spectroscopy, its unique advantages are also significant, i.e. the ultrahigh line density of above 20,000 lines/mm formed over wafer scale substrate, and the high diffraction orders enabled by the multilayer, together they bring the unprecedented angular dispersion that is vital for many applications, including the high resolution spectroscopy, and others like a compact spectrometer for space and lab-based applications, X-ray beam splitter, pulse shaper, and so on. According to the comments, we made following revisions:

- (1) We have changed the title to “Realization of wafer-scale nanogratings with sub-50 nm period through vacancy epitaxy” by removing the “ultrahigh resolution”, and changed that sentence at the beginning of the conclusion part to “we demonstrated a new method that can be used for the fabrication of high line density multilayer gratings with over 20,000 lines/mm working at the EUV and soft X-ray region”. We shifted the emphasis from “realization of high resolution

gratings” to “demonstration of a new method to fabricate ultrahigh line density gratings that can be potentially applied for high resolution spectroscopy and many other applications” throughout the manuscript.

- (2) Although the fabricated grating profiles are close to the sinusoidal shape rather than triangular ones due to the oxidation of GaAs surface, the efficiency of the multilayer nanograting with the real substrate profile obtained from the XTEM image is not much lower than that of the triangular one. For example, the Cr/C multilayer grating described in the paper shows a theoretical efficiency of 3% at ~260 eV using the real substrate profile, while that of the triangular one is 4%, with a slight shift of the peak energy, as shown in Figure R1 below. This can be taken into account in the future optimization. **This discussion has been added in the last paragraph of the section “design and fabrication of multilayer nanogratings”, page 7.**

Figure R1: The simulated diffraction efficiency of the Cr/C multilayer grating with the real substrate profile from TEM images and the triangular profile.

- (3) The better blazed profile obtained from miscut crystals is one way to improve the efficiency and will be investigated in further experiments. The period error of the gratings fabricated on GaAs surfaces with and without miscut can be reduced by using several methods from different aspects as described in the reply to the 2nd comment. These engineering issues require further systematic studies, which we think is beyond the scope of this paper. On the other hand, the efficiency can also be improved by depositing the multilayer mainly on one side of the facets (blaze facets). In this case, the absorption from the multilayer on the other side of the facets (antiblaze facets) is canceled and such a grating can achieve close to the maximum efficiency of the ideal blazed grating (antiblaze angle=90 °), as formerly discussed in [27]. This

asymmetric coating can be explored via the glancing deposition technique and the detailed process will be studied in the future. **This discussion has been added at the end of the 1st paragraph of the section “design and fabrication of multilayer nanogratings”, page 7.**

Reference:

[27] Yang X, Kozhevnikov I V, Huang Q, Wang Z. Unified analytical theory of single-order soft x-ray multilayer gratings. *J Opt Soc Am B* **2015**, 32(4): 506-522.

2. **Authors claim that “the diffraction efficiency and ultrahigh-resolution of these multilayer gratings were demonstrated.” However, GISAXS measurements showed +/-5% low-frequency variations of the grating period (Fig. 5c). This is not a small number for high resolution applications. For example, gratings for high resolution spectrometers (such as one in Ref. [31,32]) often have variable groove density which gradually changes according to a specific law by +/-5% or so over the whole grating length. The random low frequency variations of the groove density should be at least an order of magnitude smaller (say 0.5%) otherwise the imaging properties and hence the resolving power of the spectrometer will be compromised. Moreover, the reported 5% errors in groove period were extracted from the GISAXS measurements performed along the grooves, when a relatively small number of the grooves was illuminated (due to a small beam footprint in the transverse direction). It would be more correct to evaluate the periodicity errors from the q-perpendicular data for $\phi=0^\circ$ geometry (Fig. 5d). Since all the grooves are illuminated in this geometry the errors might be found much higher than 5%. In the view of the mentioned above the authors’ statements such as “we successfully demonstrated a new method for the fabrication of ultrahigh-resolution EUV and soft X-ray diffraction gratings” or “the diffraction efficiency and ultrahigh-resolution of these multilayer gratings were demonstrated“ seem not supported by the results reported.**

Response: Thanks for the reviewer’s comments. We have further analyzed the regularity of the grating line density by performing new GISAXS and AFM measurements and also analyzed it from the peak width of the detector scan of the soft X-ray diffracted beam.

There is some misunderstanding of the $\pm 5\%$ error presented in the old GISAXS figure and we did

not explain it well. This is the uncertainty of the estimated grating period of 40 nm from GISAXS, which is caused by several factors: the real period variation, instrumental factors like the geometrical settings, beam size and divergence, and grating orientation (Φ) with respect to the X-ray beam. The largest error is the Φ angle error in the last GISAXS measurement. For this, we have repeated the measurement on the grating sample and made an improved alignment of the Φ angle at $\Phi=90^\circ$ (X-ray beam parallel to the grating lines) with more careful analysis. The X-ray beam size on the sample is 0.4mm (beam width) x 10mm (projected beam length on the sample). The new results are shown in Figure 5 and the following Figure R2, which result in a grating period of 41 nm with an uncertainty error of $\pm 2\%$ now. This means the real period variation would be smaller than $\pm 2\%$ for the illuminated area. We also performed GISAXS with the X-ray beam perpendicular to the grating lines ($\Phi=0^\circ$). Nevertheless, due to geometrical limitations of the scattering geometry at $\Phi \sim 0^\circ$, the relatively weak scattered intensity at large incidence angles ($\alpha_i > 2^\circ$) necessary to observe grating peaks in forward scattering direction (particularly for the ultrashort period gratings), and the relatively weak X-ray source intensity, we were not able to observe grating peaks at $\Phi=0^\circ$. This is not related to the grating period accuracy.

Figure R2: GISAXS patterns of the nanograting structure on the GaAs sample measured with higher accuracy at (a) $\Phi=0^\circ$, grating lines perpendicular to the X-ray beam and (b) at $\Phi=90^\circ$, grating lines parallel to the X-ray beam.

AFM measurements with scanning areas of $3 \times 3 \mu\text{m}^2$ (covering ~ 60 periods) were performed at different positions on the grating sample. Based on the main peak width (FWHM) of the power spectral density function of the AFM images as shown in Figure R3, the variation of grating period in the local areas are in the range of $\pm 1.2\% \sim \pm 2.3\%$. The smallest variation is close to the

suggested value of $\pm 0.5\%$. Meanwhile, according to the angular width (FWHM) of the measured diffraction peak as shown in Figure 4(c), we estimated that the period variation is $\pm 2\%$, after subtracting the other broadening effects from the beam divergence, energy bandwidth, beam size and detector aperture. The beam size on the sample is around 0.4 mm. The period variations determined from different methods are consistent with each other. **These analyses have been added in the end of the last paragraph of the section “characterization of diffraction efficiency and angular dispersion”, page 10~11.**

Figure R3: One dimensional power spectral density function of one of the AFM images of the multilayer grating.

For the sample of 2 inch size, a small gradual drift of the grating period from the center to the edge is observed which is $\pm 2.1\%$ according to the shifted peak positions of the measured 1st diffraction order (Figure 5d). This can be caused by the inhomogeneity of the ion beam flux and the temperature gradient on the sample holder. The SEM measurements at different positions of the grating wafer also confirmed this variation at the same level. **These discussions have been added in the end of the section of “Multilayer nanograting on the wafer scale”, page 12.**

Despite the residual variation of the period, it should be noted that the high precision requirement of e.g. $\pm 0.5\%$ error of the ultra-small grating period ($D=50\text{nm}$) means an extremely small absolute error of $\pm 0.25\text{nm}$. This strict requirement from high line density/high resolution gratings is a great technical challenge for all existing grating fabrication technique. To achieve this precision, further systematic engineering and theoretical work will be performed considering the following factors:

- a. Substrate quality: Use the GaAs substrate with higher alignment accuracy of the crystalline plane, less lattice defects and impurities, and better surface figure accuracy.
- b. Ion irradiation fluence: The number of structure defects on the nanograting surface can be

reduced by optimizing the fluence and erosion time.

- c. Ion beam quality: Use a more homogeneous ion beam with a larger diameter. In this case, the homogeneous irradiation area can cover the whole grating area.
- d. Temperature control: Improve the thermal homogeneity of the vacancy epitaxy process by using a high quality heating plate with reduced temperature gradient.
- e. Theoretical modeling: further study the vacancy epitaxy process with dynamic modeling to better understand the periodic self-assembly mechanism and find other possible factors that can result in the period variation.

These discussion have been added in the 1st paragraph of the “Discussion” section, page 12~13.

On the other hand, the developed nanogratings can also be applied for many other applications where a large angular dispersion is desired rather than the high resolution and high precision of the structures. For instance, (a) the symmetric nanograting working at the conical diffraction mode can be used as a beam splitter for photon diagnostics and pump-probe experiments required in the free electron laser facilities³⁵. (b) It can be used as the key dispersion optics in an X-ray pulse shaper system to further compress the FEL pulse into sub-femtosecond level and fully control its spectro-temporal properties which is the future development direction of FEL^{35,36}. (c) The nanograting can also be used for the EUV and soft X-ray telescope where the compactness and light weight of the spectrometer are very critical for a space-based mission. **These discussions have been added in the end of “Discussion” section, page 13.**

References:

[35] David Gauthier et al., Chirped pulse amplification in an extreme-ultraviolet free-electron laser, Nat. Commun. **2016**, *7*: 13688.

[36] A. Aquila, M. Drescher, T. Laarmann, M. Barthelmeß, H. N. Chapman, and S. Bajt, Moving the Frontier of Quantum Control into the Soft X-Ray Spectrum, International Journal of Optics 2011, **2011**, 417075.

3. The poor shape of the grating grooves and high-frequency errors in grating periodicity are consistent to the low diffraction efficiency of the multilayer coated gratings. For example, diffraction efficiency of about 10% was measured for the Mo/Si coated grating

while efficiency of an ideal grating is expected to be close to the reflectance of the multilayer which is about 60% (according to http://henke.lbl.gov/optical_constants/). Even more drastic mismatch between efficiency of 1% (Fig. 3f) and expected reflectance of about 15% of a Cr/C multilayer is observed.

Response: In theory, the symmetric shape of the grating groove indeed results in a relatively low efficiency compared to the multilayer reflectance. This can be improved by changing the symmetric shape into asymmetric one by using a miscut wafer in the process of vacancy epitaxy, or trying to coat the multilayer mainly at one side of the grating facets through e.g. glancing deposition technique, as mentioned in the response to the 1st comment. These methods are currently under investigation. Taking into account the smoothing effect of the groove shape, the experimental efficiency is actually close to the expected value. For example, the measured reflectance of Mo/Si reference multilayer (20 bilayers) is 46%, which is 71% of the ideal reflectance. Assuming the real multilayer imperfections have the same effect on the ML grating efficiency, and using the real groove profile after coating, the expected efficiency of the Mo/Si multilayer grating is 25% (theoretical value shown in Figure 3d)*71%=17.8%, which is not much higher than the experimental value of 11%. **This comparison has been added in the part of the Mo/Si MBG in the section “Characterization of the diffraction efficiency and angular dispersion”, page 9.**

On the other hand, the symmetric shape of the grating is demanded for some other applications which don't require high efficiency, e.g. a beam splitter. In this case, our symmetric nanograting (working at conical diffraction mode) can generate two diffracted beams ($\pm 1^{\text{st}}$ orders) with identical intensity and one transmitted beam (0^{th} order), with very large separation. This can be applied in photon diagnostics and pump-probe experiments in free-electron laser facilities. **This discussion is added in the end of the Discussion part, page 13.**

Reviewer 2

- 1. This appears to be the first report on vacancy epitaxy. If so, atomic resolution TEM images to show the vacancy formation and reorganization should be provided in order to confirm the mechanism.**

Response: Thanks for the reviewer's comment to our work. Compared with the self-organization

mechanism of atom by atom in the conventional MBE method, low-energy ion irradiation at elevated temperature generate vacancies on the GaAs surface, and these vacancies will self assemble according to the crystalline symmetry of the surface. In practice, during the ion erosion process, the structure of the nanopattern evolves continuously from disorder to highly order. It is very challenging to directly show the vacancies dynamics by atomic resolution TEM. In-situ TEM during low-energy ion irradiation would be needed.

2. Surface smoothness: the authors place that part in the supplemental information. I would recommend moving that to the main text, because it is important to address surface scattering for optical gratings.

Response: Thanks for the reviewer's suggestion. The smoothing effect of the Mo/Si multilayer coating on the grating profile has been moved from the supplement material to the section of “characterization of grating efficiency”, and Figure S2 has been partially merged in Figure 3. Related text has been added in the description of Figure 3 (page 8) as “To further study the effect of the smoothed grating profile on efficiency, the blaze angles after different number of bilayers deposited were estimated from the TEM image (Figure 3e). The theoretical efficiency of the Mo/Si multilayer grating with the reduced blaze angles were calculated and shown in Figure 3(f). As the blaze angle decreases from 18.65° (bottom area) to 14.27° (middle area) and 10.82° (top area), the 2nd order efficiency drops continuously while the 1st order efficiency increases. Evidently, the grating with the smaller blaze angle is more matched with the multilayer structure at the 1st order than the 2nd order which transfers the energy mainly to the 1st order.”

3. The multilayer Mo/Si and Cr/C deposition condition on top of the GaAs grating structure needs to be described clearly and the effect of the GaAs surface smoothness on the layers above need to be mentioned.

Response: The detailed deposition conditions are described in the Methods section, and a new reference of the fabrication of a high reflectance Cr/C multilayer is added as [37]. Due to the very small layer thickness, the growth of X-ray multilayer requires super-smooth substrates with a typical roughness of $<0.3\text{nm}$ (RMS). The surface roughness of the GaAs grating facets is around 0.2 nm as measured by AFM, which is sufficient for the growth of multilayer. This description is

added in the 1st paragraph of the section “formation of nanograting pattern on GaAs (001) surfaces”, page 5.

Reference:

[37] MingwuWen, Li Jiang, Zhong Zhang, Qiushi Huang, Zhanshan Wang, Rui She, Hua Feng , HongchangWang, High reflectance Cr/C multilayer at 250 eV for soft X-ray polarimetry, Thin Solid Films 2015, **592**: 262–265.

4. In terms of the impact of this work, what is the fundamental limit of the range of grating periodicity using this method? How general is this method? Other than GaAs, would this method apply to other compound semiconductors with zinc-blende structure with aligned dimer surface? Since this is going to be used to soft X-ray optics, is GaAs hardy enough for extended exposure?

Response: Thanks for the reviewer’s important question. The vacancy epitaxy method can be applied to many semiconductor surfaces and will create different nanopatterns depending on the symmetry of the surface. For instance, vacancy epitaxy on InAs surfaces, another compound semiconductor with zinc-blende structure and two-fold symmetry, can also generate a similar nanograting structure as shown in Figure R4. The period of the grating depends on surface symmetry and irradiation temperature. For instance, on a GaAs(100) surface, the range of grating period can be tuned from 40nm to 90nm as the irradiation temperature increased from 350 °C to 550 °C, due to the fact that vacancy diffusion is enhanced at elevated temperature. As the process on other compound semiconductors with zinc-blende structure and aligned dimer surface has different temperature windows and period, the range of grating period can also be extended by employing different III-V materials.

In addition, the GaAs grating structure remains stable below 500°C whose thermal stability is even higher than that of the the multilayer. For a spectrometer system, the soft X-ray intensity emitted from the sample is also weak and will not cause irradiation damages on the optics. Therefore, GaAs nanogratings are hard enough to be applied in the spectrometer system.

Figure R4: The nanogroove structure formed by vacancy epitaxy on (a) GaAs(100), (b) InAs(100).

Reviewer3

1. **The blazed gratings are important to improve the transmission of the grating by selecting the appropriate blazed angle. In the presented work it is not clear how the blazed can be selected independently from the groove density. Some comments of the coupling between the blazed angle and groove density will be welcome to better understand the strength and/or weakness of this approach with respect other methods.**

Response: Thanks for the reviewer's comment. The blaze angle can be tuned by two methods. One is using the miscut surface of GaAs. The blaze angles of the grating structure are reduced from 19° , to 9° and 4° as the vacancy epitaxy performed on vicinal GaAs(001) surfaces with miscut angles of 0° , 10° and 15° , respectively. However, the period of the structure is increased accordingly.

Alternatively, the blaze angle can be reduced by coating the grating with a thick layer to smooth out the grating profile before the deposition of multilayers. Similar effect of smoothing was seen in the growth of multilayer on the nanogratings (Figure 2e).

To achieve the resonant coupling/interaction between the incident wave and a certain order diffracted wave for maximum efficiency, the groove density, blazed angle and the d-spacing of the multilayer need to be matched together, at a fixed photon energy. This is to satisfy both the grating equation condition and the generalized Bragg condition as discussed in ref [27]. Generally, for the multilayer nanograting with a fixed line density, the blazed angle needs to be reduced to below 19° (initial angle generated by the process) as working at higher photon energy, in order to satisfy the two conditions mentioned above. Meanwhile, a smaller blazed angle leads to smaller d-spacing of the multilayer. So the blazed angle cannot be too small in order to make the d-spacing achievable.

Using a higher order diffraction allows the use of a larger blaze angle than the 1st order would require. Due to the extremely small grating period of the developed structure, the highest working energy is limited compared to conventional multilayer gratings with large periods [30], based on the above mentioned principles. For instance, assuming the smallest d-spacing of 3nm and a grating period of 50nm, the highest working energy is below 3keV. A detailed diffraction model of the ultrahigh line density multilayer grating is being built and will be presented in another work.

These descriptions are added in the section of “design and fabrication of the multilayer nanogratings”, page 6.

References:

[27] Yang X, Kozhevnikov IV, Huang Q, Wang Z. Unified analytical theory of single-order soft x-ray multilayer gratings. J. Opt. Soc. Am. B 2015, 32(4): 506-522.

[30] Senf, F, Bijkerk F, Eggenstein F, Gwalt G, Huang Q, et al., Highly efficient blazed grating with multilayer coating for tender X-ray energies. Opt. Express 24, 13220-13230 (2016).

2. The blazed grating quality are defined with the blazed angle (exposed surface) and “anti” blazed angle (unexposed surface) which are in principle highly asymmetric. It seems that the ultimate groove density grating (55nm) shows a symmetric structure which indicates the limitation of the applied method. Some comments on this limitation are requested.

Response: The vacancy epitaxy process on the GaAs (001) surface generates a nanograting with symmetric facets. This reduces the theoretical efficiency compared to an ideal blazed grating. However, several methods can be developed to overcome this issue. A miscut crystalline surface can be used to generate the asymmetric facets similar to the conventional blazed grating shape as shown in Figure1 (h). On the other hand, if the multilayer can be deposited mainly on one side of the facets (blaze facets) while the absorption from the multilayer on the other side of facets (antiblaze facets) is canceled, the maximum efficiency can still be comparable to an ideal blazed grating as discussed in [27]. This structure may be realized by using the glancing deposition technique instead of the current normal incidence deposition. **These discussions have been added at the end of the 1st paragraph of the “Design and fabrication of the multilayer nanograting” section, page 7.**

Moreover, the symmetric groove shape is demanded for some other applications except

spectroscopy, e.g. the grating beam splitter. In this case, our symmetric nanograting working at the conical diffraction mode can generate symmetric diffraction orders together with the reflected 0th order with extremely large separation, which is important for photon diagnostics and pump-probe experiments in free-electron laser facilities. **This has been added in end of the “Discussion” section, page 13~14.**

Reference:

[27] Yang X, Kozhevnikov IV, Huang Q, Wang Z. Unified analytical theory of single-order soft x-ray multilayer gratings. J. Opt. Soc. Am. B 2015, 32(4): 506-522.

3. The claimed energy resolution is based on a linear fit between photon energy and diffraction angle (see figure 4b and 4c) extrapolated to a 11 meter spectrometer. This in no means a good measurement/estimation of the ultrahigh energy resolution of the grating. We strongly recommend to rephrase this part of the manuscript including title. Moreover no comments are provided regarding the “width” of the intensity distribution (in figure 4 b and 4c) which is a clear limiting factor in the true energy resolution. Same applied on optical quality of the blank substrate (for example slop errors) which will limit the energy resolution. Some open question remains also on the remaining optical quality of the blank after the low energy ion irradiation in the proposed approach.

Response: Thanks for the reviewer’s comment. We agree with the reviewer’s suggestion and have shifted the main discussion from resolution to the angular dispersion property, given to the current experimental results. The part of the estimation of resolution has been rephrased as following. The measured angular dispersion (the separation of the peak diffraction angles of different energies) of the multilayer gratings is consistent with the theoretical value of a 20,000 lines/mm grating determined by the grating equation. So the general function of the nanograting is proved. Due to the ultrahigh line density and the high order used, the obtained angular dispersion of the fabricated multilayer grating is 4.5 to 6.3 times larger than a 5,000 lines/mm grating which is almost the highest line density possibly achieved by modern ruling technique [19]. This can provide a much larger spatial dispersion of the adjacent energies on the image plane and thus higher resolving power, assuming other technical parameters like the variation of grating period, the slope error of the grating substrate, etc., are the same.

We acknowledge that the assumption above of identical technical parameters of the new

nanogratings as modern grating optics is not fully valid yet. According to the reviewer's comment, we performed detailed analysis of the angular width of the diffraction peak. Taking the Cr/C multilayer grating as an example, the FWHM width of the 4th order diffraction peak (Figure 4c) is 3.3 times larger than that of the direct incident beam. After subtracting the broadening effects from the incident beam divergence and energy bandwidth, beam size and acceptance angle of the detector, we found that the main reason for the broad peak is the small variation of the grating period, which is estimated to be $\sim \pm 2\%$. This is indeed a limiting factor for achieving high spectral resolution. Nevertheless, reducing the period variation to a good level of e.g. $\pm 0.5\%$ for such ultrashort period gratings means an absolute error of only $\pm 0.25\text{nm}$. This is a great challenge posed by the ultrahigh resolution grating optics for every existing grating manufacture techniques. To achieve this, several engineering factors can be studied, like employing a better quality substrate with less lattice defects and impurities, further optimizing the ion irradiation parameters, improving the homogeneity of the irradiated process, and so on. **These discussions are added in the end of the section "characterization of the diffraction efficiency and angular dispersion", page 10~11, and the 1st paragraph of the "Discussion" section, page 12.**

On the other hand, the developed multilayer nanograting can not only be applied for high resolution spectroscopy, but also for the X-ray beam splitter, ultrashort pulse shaper, spectrometer in space telescopes, where the large angular dispersion and a compact size is more important. **These have been added in the end of the "Discussion" section, page 13.**

In this paper, the grating structure was fabricated on a thin wafer with no requirement on figure. Nevertheless, the polishing process of GaAs is similar to Si, and it is widely used in semiconductor industry for fabricating infrared detectors. Slope errors of sub-micron radians can be achieved on small bulk GaAs substrates which could be further optimized to the level of high precision Si substrate. The ion irradiation process was performed under near normal incidence, so the etching and self-assembly speed should be the same over the whole area assuming a uniform ion flux, which should not deteriorate the initial surface figure in principle.

4. Figure 4b shows interesting features such as "side lobes" which are not explained. Moreover those "side lobes" are mainly dominating the linear fitting between photon energy and diffraction angle, therefore it is mandatory to give a physical interpretation

of those features. In addition, the central feature of Figure 4b shows a different dependence between photon energy and diffraction angle than the one defined by the “side lobes”. This requires a more detailed understanding of the observed features.

Response: The side lobes are caused by the interference effect of the multilayer grating. The central maximum efficiency is resulted from the constructive interference of all the diffracted waves from different interfaces, while the local maximum of the side lobe is caused by the constructive interference of only part of the diffracted waves. This is a special characteristic from the multilayer structure. We did not perform linear fitting of the two dimensional diffraction measurements in Figure 4. The angular dispersion, i.e. the relation between photon energy and diffraction angle, is calculated from the shift of peak angular positions measured at different energies, within the central Bragg diffraction region. To clarify this, the dashed curve in Figure 4 (representing the energy-angle relation) is limited to the central maximum area, as also shown in Figure R5.

The resonant working energy of the multilayer grating partially depends on the grating period. As the current grating sample has a small local period variation of $\sim \pm 2\%$, the slight difference of the energy-diffraction angle dependence between the central area and the side lobe area can be caused by the slightly different grating period that is resonantly matched with the different energies. Nevertheless, the grating will not work at the side lobe area in applications due to the very low efficiency.

These discussions have been added in the 3rd paragraph of the section “characterization of diffraction efficiency and angular dispersion”, page 9-10.

Figure R5: Corrected Figure 4b.

Reviewer #2 (Remarks to the Author):

My questions and concerns have been addressed adequately and I recommend publish the revised version as is.

Reviewer #3 (Remarks to the Author):

We thank the authors for addressing carefully each of the open issues raised during the previous review. We believe that the authors managed to improve the quality of the present manuscript to a level which could be publishable.